# Optimization of High-Frequency Ultrasound Imaging to Detect Incremental Changes in Mineral Content at the Cartilage–Bone Interface Ex Vivo

**DOI:** 10.3390/biomimetics10030160

**Published:** 2025-03-05

**Authors:** Akshay Charan, Parag V. Chitnis, Caroline D. Hoemann

**Affiliations:** Department of Bioengineering, George Mason University, Manassas, VA 20110, USA; acharan2@gmu.edu (A.C.); pchitnis@gmu.edu (P.V.C.)

**Keywords:** cartilage–bone interface, high frequency ultrasound, tidemark, decalcification, hard/soft interface remodeling, non-decalcified histology, von Kossa stain

## Abstract

(1) Background: Osteoarthritis is a degenerative disease of the whole joint marked by cartilage–bone interface (CBI) remodeling, but methods to monitor subtle changes in mineralization are lacking. We optimized a non-destructive ultrasound imaging method to monitor incremental shifts in mineralization, using brief decalcification as a mimetic of CBI remodeling. (2) Methods: We used a 35-MHz transducer to scan 3 mm diameter bovine osteochondral explants wrapped with parafilm to produce surface-directed decalcification and dedicated 3D-printed holders to maintain sample orientation. Customized MATLAB codes and a matched pair design were used for quantitative hypothesis testing. (3) Results: Optimal scan precision was obtained when the High-Frequency Ultrasound (HFUS) focal distance was trained at the CBI. HFUS cartilage thickness increased by 53 ± 21 µm or 97 ± 28 µm after three or seven hours of ethylene diamine tetra-acetic acid (EDTA) (but not PBS), respectively, and was highly correlated with histological cartilage thickness (R = 0.98). The en face CBI backscatter pattern was irregular and shifted after the EDTA-displacement of the mineral front. Collective data suggested that the −10 dB echogenic CBI signal originated from the mineral front and varied topographically with undulating mineral thickness. (4) Conclusions: This imaging approach could be used to monitor tidemark remodeling in live explant cultures, toward identifying new treatments that inhibit tidemark advancement and slow osteoarthritis progression.

## 1. Introduction

Osteoarthritis (OA) is a leading cause of joint disability, affecting approximately 7.6% of the global population in 2020 [1] and imposing significant physical, psychological, and economic burdens on society. OA is primarily characterized by the progressive degeneration of articular cartilage and pathological changes in the subchondral bone and surrounding tissues, leading to joint pain, stiffness, and impaired mobility [2,3,4] While much research has focused on characterizing the mechanisms of articular cartilage breakdown, the underlying changes at the cartilage–bone interface (CBI) represent a critical yet underexplored aspect of OA pathophysiology [5,6]. Toward the end of skeletal maturity, a ~5 µm thick tidemark line develops at the mineral front of the calcified cartilage layer [7]. The tidemark is believed to serve as a functional barrier to endochondral ossification, a process of normal fetal development whereby growth cartilage mineralizes and becomes replaced by bone [8]. During OA progression, the cartilage deep zone shows signs of renewed endochondral ossification including irregular creeping mineralization, tidemark duplication, and punctate vascular invasion [9,10,11,12,13]. Traumatic joint injury can initiate tidemark remodeling but the mechanisms remain poorly understood. This study aimed to develop a non-destructive imaging technique to measure incremental changes in mineralization at the CBI, as a step toward creating a method to monitor “live” tidemark remodeling in an explant model system.

High-frequency ultrasound (HFUS) is a non-destructive method for analyzing explanted osteochondral joint tissue that offers higher resolution than magnetic resonance imaging (MRI) and correlates well with histological evaluations [14,15,16,17]. MRI produces a signal at the CBI, but the signal was located above the tidemark when the 3-D images were co-registered with micro-CT scans [18]. When the HFUS transducer is pointed at the articular surface, strong echogenic signals are generated from the articular surface and CBI [15,19]. Assuming a constant value of the speed of sound through cartilage, in the range of 1590 m/s to 1660 m/s [20], HFUS can be used to determine cartilage thickness as the difference between the location of these two peaks as a quantitative ultrasound (QUS) parameter. HFUS was previously used to measure cartilage thickness in equine metacarpal, bovine knees, hips, or patella, porcine shoulders or elbows, normal human condyles, OA human lateral tibial plateau or knee specimens using 9 to 50 MHz transducers [10,15,20,21,22,23,24,25], and in rat distal femurs with 50 or 80 MHz transducers [17,26]. These prior studies were unable to convincingly distinguish whether the second backscatter peak arises from the trabecular bone or the tidemark line of the CBI. Huang et al. [10] co-registered HFUS and micro-CT images of OA osteochondral specimens and noted discrepancies between the micro-CT tidemark surface and HFUS CBI surface for reasons that remain unclear. A key limitation of these prior works is that the ultrasound parameters such as focal length and frequency, which determine imaging depth and resolution, were not standardized or optimized. Few studies have explored the effect of focal depth on the reproducibility of QUS parameters or image quality. For example, Passmann and Emert previously analyzed the effect of HFUS focal depth (30–130 MHz) on B-mode image quality in skin specimens [27]. The authors noted that a penetration depth of 1 to 3 mm is required for most medical applications and found that the best contrast was achieved with depth-focus scanning. Fukumoto et al. [28] demonstrated that adjusting the ultrasound focal depth to match the region of interest (ROI) improves the accuracy of echo intensity as a measure of muscle quality by reducing attenuation effects in B-mode images of human thigh muscles. Similarly, Duggan et al. found that in lung ultrasound, B-lines were best visualized using a curvilinear transducer with the focal position at the pleural line, resulting in better image quality [29]. In addition, previous studies of osteochondral tissues were lacking important gold standards such as non-decalcified histology to assess cartilage thickness and visualize mineral deposits.

The purpose of this study was to determine the role of the mineral front in HFUS backscatter at the CBI. We hypothesized that HFUS cartilage thickness increases after incremental decalcification of the calcified cartilage layer. Although higher frequencies offer better scan resolution, it is well known that tissue penetration becomes attenuated as frequency increases [30]. Therefore, we compared HFUS scans of bovine explants obtained using two frequencies, 15 MHz and 35 MHz, because frequencies above 35 MHz would likely not provide adequate penetration in tissue to assess the CBI. Our initial studies showed that both the 15 MHz transducer with a focal distance of 50.8 mm and a 35 MHz transducer with a focal distance of 12.7 mm could measure cartilage thickness in bovine specimens, but the 35 MHz transducer with its superior resolution was needed to detect incremental changes in mineral content at the CBI. Using the 35 MHz transducer, we systematically optimized focal depth, explant sample preparation, and sample orientation during the scan with bovine osteochondral explants. We also verified the accuracy of our HFUS cartilage thickness measures using non-decalcified histology as the ground truth.

## 2. Materials and Methods

### 2.1. Osteochondral Explant Preparation

This study used fresh intact bovine shoulders obtained from a local butcher within 24 h of slaughter and confirmed them as never frozen (*n* = 4 adult, *n* = 1 veal). Our study was carried out according to the 3 R’s (Reduction, Refinement, Replacement) in The Principles of Humane Experimental Technique [31] by using agricultural specimens as an alternative to live animal experimentation in research. An adult shoulder specimen was defined by a ≥7.5 cm humeral head width. The veal humeral head was 5.4 cm wide. The humerus was immobilized in a vice then the humeral head was exposed by dissection so that cylindrical osteochondral explants (3 mm in diameter, 6 to 10 mm length) could be extracted from the central region using 8G Jamshidi needles (McKesson, Richmond, VA, USA, Product #DJ4008X, or CareFusion, Vernon Hills, IL, USA, Product #TJC4008, Appendix A). Each explant was ejected from the Jamshidi needle by pushing on the bony side upward so that the cartilage was not damaged. Explants from the weight-bearing area of the shoulder had visibly thinner cartilage than the explants collected closer to the periphery (around 0.5 mm difference). Cartilage surfaces were kept humid with periodic irrigation with sterile phosphate-buffered saline (PBS, Sigma-Aldrich, St. Louis, MO, USA). Explants were trimmed of bone with a razor if necessary to 10 mm length, imaged with a digital camera with a ruler for scale, and stored at −20 °C in humid chambers (cryovials containing a PBS-soaked Kimwipe at the base) to preserve the structural integrity of cartilage and to prevent enzymatic degradation. Explants for the final decalcification study were generated by two team members: one to carefully position the needle perpendicular to the cartilage surface and the other to extract the core with the Jamshidi needle. Samples incubated in EDTA or PBS were systematically selected from different regions of each shoulder.

### 2.2. Pilot Study with a 15 MHz Transducer

Six adult bovine explants were scanned in a PBS bath with a 15 MHz transducer (Olympus A319S-S, Olympus NDT, Westborough, MA, USA, optical focal resolution 50.8 mm and time of flight, TOF, 67.7 µs) after extraction, then re-scanned after formalin fixation, and then again after 1, 2, and 4 days of demineralization in 10% *w*/*v* EDTA in PBS (Sigma-Aldrich, pH 7.4). Samples were scanned in custom 3D-printed holders but separated from the holder during fixation and decalcification. A custom MATLAB code was used to collect 9 scan lines from the center of each explant. HFUS cartilage thickness was measured as the spatial distance between the 2 major peaks, assuming a 1600 m/s speed of sound through cartilage (Appendix A). HFUS cartilage thickness measures were variable and slightly lower after formalin fixation and not significantly changed after 1 day of EDTA treatment, which decalcified 263 μm through the subchondral bone plate (Table 1, Appendix A). Therefore, all further studies used a 35 MHz transducer with a ~23 μm axial resolution (instead of the 15-MHz transducer that provided ~53 μm axial resolution, Appendix A). Based on these results, all further studies used unfixed explants to avoid the formalin-induced tissue shrinkage artifact and kept each explant in a dedicated 3D-printed holder between repeat scans to preserve sample orientation (Table 1).

### 2.3. HFUS Scanning with a 35 MHz Transducer

The custom HFUS system employed a 35 MHz transducer (Olympus P135-2, R = 0.50″, Part #200591, Olympus NDT, Westborough, MA, USA, focal depth 12.7 mm and round-trip time-of-flight (TOF) 16.9 µs, assuming a 1500 m/s speed of sound in PBS) mounted on a two-axes motorized stage programmed to raster scan an area of 6 mm × 6 mm with a 100 µm step size at 0.5 mm/s (Figure 1, Appendix A). To ensure consistent positioning of the explant before and after decalcification, we designed U-shaped 3D-printed holders with a 3.2 mm hole in the center into which we press-fit the explant. The holder was designed so that it could be placed in and removed from a custom 3D-printed support fixed to the base of the watch glass with double-sided tape (Figure 1B,C). In the final decalcification study, the explant was fit in the 3D-printed support, then a thin strip of parafilm was wrapped around the circumference before the first scan to permit only “surface-directed” decalcification. Parafilm wrapping ensured that decalcification was restricted to the targeted region, minimizing the risk of unwanted outside-in decalcification at the CBI for accurate imaging. After submerging in degassed PBS, the explant was centered in a 6 × 6 mm scanning area, and the depth of focus was aligned by an adjustable base so that the 16 µs time of flight (TOF) was trained at the cartilage surface (the first peak), mid-zone, or the CBI (second peak, see Table 1). After the first scan, the explant in the 3D-printed holder was then transferred to a watch glass filled with 200 mL 10% *w*/*v* EDTA in PBS (pH 7.2) completely submerging the explant or PBS. The first decalcification study exposed specimens to EDTA for 1.5 h, 3 h, 4 h, and 22 h (veal) or 3 h, 7 h, and 48 h (adult shoulder 2) and the second decalcification study incubated specimens for 3 h or 48 h (EDTA or PBS negative control, adult shoulder 3) or for 7 h or 96 h (EDTA or PBS, adult shoulder 4). The justification for testing 3 h or 7 h of EDTA was based on our goal to decalcify for a minimal period of time needed to generate an incremental shift in the tidemark. The 48 h and 96 h EDTA treatments were used as positive controls for complete decalcification and PBS incubations were used as negative controls for tissue swelling. The treatment solution was agitated with a stir bar in a corner of the dish throughout the treatment period to ensure dynamic decalcification at room temperature. Following decalcification, samples were washed with PBS for 15 min to remove residual EDTA before re-scanning (Table 1).

### 2.4. Data Acquisition, Image Processing, and Data Analysis

Raster scanning and A-mode data acquisition at each scan step were performed using a custom instrument-control script developed using LabVIEW. An ultrasound pulser–receiver (Olympus 5073PR, Olympus NDT, Waltham, MA, USA) and oscilloscope (Lecroy HD04024A, Teledyne Lecroy Chestnut Ridge, NY, USA) were triggered using the Transistor–Transistor Logic (TTL) signal, which is a binary signal with well-defined voltage levels and is used to initiate the oscilloscope sweep and ensure that the waveform is captured at a specific point in the digital signal, thereby minimizing the waveform from jittering or drifting. The TTL signal was produced by the controller for the motorized stages for automatic synchronization and precise collection of A-mode data while the transducer was in continuous bi-direction motion to cover the 6 mm × 6 mm grid in under 20 min. A step size of 100 µm produced 61 data matrices, each containing 61 A-mode signals, to provide the requisite 3D scan, where the third dimension can be inferred by the TOF of the pulse-echo signal. The data acquisition had a sampling frequency of 500 MHz within a 10-µs time window. The pulser–receiver settings were as follows: Energy: 4; Gain: 26 dB; Damping: 2; filter: 5-MHz high pass filter (HPF). The 35-MHz transducer has a bandwidth of roughly 27–43 MHz; therefore, low-frequency signals below 5 MHz do not contribute to tissue characterization as these are simply associated with extraneous noise. Therefore, we used a 5 MHz HPF to eliminate any low-frequency noise. Raw data were processed using custom MATLAB (version R2023b, Mathworks, Natick, MA, USA) scripts. HFUS data were processed using a Hilbert transformation to find the signal envelope, and peak detection was used to identify the maximum value of the RF envelope data that was then normalized to a decibel (dB) scale. The signal originating from the articular cartilage (AC) and CBI in each scan line was identified from the two most prominent peaks. Between 400 to 800 scan lines per explant were selected for QUS measures using only scan lines with at least two peaks meeting the 50 mV threshold as a noise floor. To filter out occasional noise in the cartilage mid-zone, we collected the average amplitude of peaks that fell within a 1 µs depth range covering the two most prominent peaks. Data were used to measure Delta (mm): distance between the articular cartilage surface peak and the CBI peak assuming a constant speed of sound through cartilage (1600 m/s) [20]; Alpha (dB/mm): the slope that is measured by the difference in backscatter intensity between the CBI peak and 200 µm prior to the CBI peak where high values indicate a sharp increase in backscatter at the mineral front and low values indicate a significant loss of mineral content at the CBI; and CBI Backscatter Intensity (dB): the amplitude of the CBI backscatter peak. Scan data were used to reconstruct transverse images from the middle of the explant, segmented en face images of the articular cartilage surface (included 200 µm beyond the first peak), and CBI (included 200 µm above and 200 µm below the second most prominent backscatter peak), as well as 3D images using the Maximum Intensity Projection (MIP) algorithm. The MIP algorithm is a visualization technique that projects the highest intensity value along each depth axis onto a 2D plane. In our study, MIP was used to create a 2D projection of 3D reconstruction volume of our specimens obtained from HFUS scans by displaying the strongest backscatter intensity at each depth, effectively highlighting mineralized tissue structures and their changes after decalcification.

### 2.5. Histology and Histomorphometry

After scanning, explants were snap-frozen in optimal cutting temperature (OCT) compound using liquid nitrogen and stored at −20 °C. To optimally preserve bone, bone marrow, and cartilage integrity, transverse 20 µm thick cryosections were collected using a Leica cryostat by adhering a piece of scotch tape to the cryoblock before cutting the section and then securing the taped section face-up on a clean microscope slide. Sections were stored at −20 °C. Von Kossa staining was carried out on cryosections thawed to room temperature, rinsed in double-distilled water for 5 min to remove OCT, treated with 5% *w*/*v* silver nitrate in ddH20, exposed to UV light for 30 min., rinsed in ddH20, treated with 3% *w*/*v* sodium thiosulfate for 3 s to fix the stain, and then flooded with 10-fold diluted filtered Harris Hematoxylin for 5 min. Sections were rinsed with tap water, mounted in aqueous medium, and 2.5× magnification digital images were captured with a Zeiss Axiovert A25 microscope and ZEN 2.3 (czi image) software (Carl Zeiss Microscopy, GmbH, 2011, Jena, Germany). Non-calcified cartilage thickness was measured using Image J using a microscope scale bar to calibrate pixels/µm. Ten line measures from the cartilage surface to the mineral front were collected across the entire explant width and averaged.

### 2.6. Statistical Analysis

Statistical analyses were conducted using JMP software (JMP Pro 17.2.0, JMP Statistical Discovery, LLC, Cary, NC, USA). Differences between baseline and rescan (N = 30), post-decalcification (N = 6, 3 h or 7 h EDTA, or N = 3, 48 h or 96 h EDTA), or post-PBS (N = 3) QUS measurements were assessed using matched paired *t*-tests. Variations in QUS parameter changes across different conditions were analyzed using one-way ANOVA, followed by post-hoc tests (Dunnett’s test and Student’s *t*-test), with significance set at *p* < 0.05. The Pearson correlation coefficient was used to evaluate the correlation between HFUS-derived cartilage thickness (Delta) and histological non-calcified cartilage tissue thickness (N = 12).

## 3. Results

### 3.1. Optimization of HFUS Focal Depth and Sample Preparation

We analyzed the effect of focal depth on 35 MHz image quality and HFUS cartilage thickness measures. Four veal and four adult explants were press-fit in asymmetric 3D-printed holders and kept in the holder while scanning at the cartilage surface, the mid-zone, and then the CBI and then they were removed from the scan bath and reinserted for three more scans at each focal depth to control for sample handling on thickness measures (Figure 1A–C). By comparing the transverse, segmented en face, and 3D images reconstructed from scans at different focal depths, we found that the backscatter signal strength shifted from the cartilage surface to the CBI as the acoustic focal distance was displaced from the cartilage surface to the CBI (Figure 2).

Veal and adult cartilage had an average HFUS cartilage thickness of 1.48 ± 0.22 mm and 0.88 ± 0.21 mm, respectively (N = 4, Appendix A). Variability in thickness between explants was partly due to cartilage being visibly thinner in the central load-bearing area than the periphery of the humeral head and inclusion of samples from both areas of the joint. When the transverse HFUS images were analyzed alongside the HFUS cartilage thickness measures, we noticed an artifact related to sample shape. The three explants with flat surfaces had the same HFUS cartilage thickness at all three focal depths and when the re-scan was subtracted from the baseline scan, we saw a difference of 20 ± 21 µm for explants with flat surfaces. By comparison, five explants with “tilted” surfaces showed a drift in average HFUS cartilage thickness at the different focal depths and a three-fold lower precision between the baseline and re-scan (69 ± 57 µm, Appendix A). We attributed the lower precision due to variable cartilage thickness in the scanning axis for oblique-shaped samples. Based on these results, additional care was taken to extract explants with flat surfaces perpendicular to the scanning axis. Subsequent scans were carried out, focusing the transducer at the CBI.

### 3.2. Optimization of the EDTA Decalcification Method

In the next experiment, we optimized the decalcification method needed to incrementally displace the mineral front. EDTA treatment of a veal explant (4 h) or adult explant (7 h) was sufficient to demineralize ~100–150 µm of calcified cartilage tissue in the middle of the explant according to histology measures. We measured a ~70 µm increase in HFUS cartilage thickness relative to the baseline in the center of the explant, but at the edges of the explant, an unwanted “outside-in” decalcification was found to take place leading to high and variable HFUS cartilage thickness around the explant periphery (dashed arrows, Figure 3, Appendix A).

To obtain “surface-directed” decalcification, we tested the effect of wrapping parafilm tightly around the explant CBI prior to HFUS scanning and EDTA treatment (Figure 1D). With this approach, we found that HFUS cartilage thickness increased by 40 μm or 70 μm, respectively, after 3 h or 7 h EDTA exposure, and by histology, decalcified only the calcified cartilage layer while preventing outside-in decalcification (unpublished observations, Pilot study, Table 1).

### 3.3. HFUS Cartilage Thickness Is Altered by Brief Decalcification and Correlates with Non-Decalcified Histology Cartilage Thickness

With the final optimized protocol, explants from two distinct adult bovine shoulders were submitted to two baseline scans and then another scan after incubation in EDTA or PBS with the 35 MHz transducer. To control for the effect of sample handling, samples in dedicated 3D-printed holders were removed from and replaced in the scan setup between the two baseline scans using tweezers to handle only the holder without touching the explant. The HFUS cartilage thickness was 1072 ± 209 µm on average with a precision of 2 µm ± 11 µm between the two baseline scans for all 30 explants. Explants gained 53 ± 21 µm in HFUS cartilage thickness after 3 h of surface-directed EDTA (0.96 mm baseline vs. 1.02 mm post-EDTA, *p* < 0.05, N = 6, Figure 4A) and 97 ± 28 µm after 7 h of EDTA (1.09 mm baseline vs. 1.19 mm post-EDTA, *p* < 0.05, N = 6, Figure 4B). The incubation of explants for 3 h or 7 h in PBS failed to increase HFUS cartilage thickness, suggesting that the observed EDTA-induced increase in HFUS thickness was not due to cartilage swelling (Figure 4A,B). The incubation of explants for 96 h in PBS, however, produced a significant 102 ± 33 µm increase in HFUS cartilage thickness over the baseline, suggesting that long-term incubation of unfixed tissue in PBS led to minor cartilage swelling (Figure 4B). In fully decalcified explants, the HFUS “cartilage thickness” increased by 425 ± 76 µm after 48 h of EDTA and 438 ± 80 µm after 96 h of EDTA (Figure 4A,B).

Histology-based cartilage thickness closely matched HFUS cartilage thickness after brief EDTA treatment, with an average difference of approximately 7 µm (average cartilage thickness: 1.229 mm HFUS vs. 1.236 mm histology, N = 12). Pearson’s correlation showed a relationship of HFUS thickness = 0.019 + 0.99 × histology thickness (*p* < 0.0001, R = 0.98, R^2^ = 0.97, Appendix A). This close agreement suggested that the second backscatter peak originated from the mineral front of the incrementally decalcified specimens.

CBI backscatter intensity is defined as the amplitude (in dB) of the second prominent peak corresponding to the CBI. It was hypothesized that incremental decalcification of the CBI would reduce the echogenicity of the CBI; however, baseline scans showed approximately −10.5 dB ± 0.7 dB backscatter energy that was unchanged after brief EDTA or PBS incubation (≤0.33 dB difference from baseline, Figure 4C,D). CBI backscatter dropped to −17.9 dB ± 0.9 dB after full decalcification (mean decrease from baseline: −6.63 dB, *p* < 0.01, Figure 4C,D). These findings suggested that the calcified cartilage layer lost its tissue density and inherent acoustic impedance after being demineralized by brief EDTA (Appendix A). The change in speed of sound will be small in comparison to the change in density upon decalcification.

Campbell et al. previously reported a dramatic shift in mechanical stiffness at the deep cartilage radial zone and tidemark by atomic force microscopy [32]. To analyze this zone by HFUS, we used Alpha as a QUS parameter to measure the upswing in acoustic impedance between 0.2 mm before the CBI and the CBI (dB/mm). Baseline Alpha (173.8 or 179.8 dB/mm) was significantly reduced after 48 h or 96 h EDTA treatment (73.5 dB/mm or 59.9 dB/mm, respectively, N = 3, Appendix A). Incubation in PBS or brief EDTA, however, had no significant effects on Alpha (178.5 dB/mm or 185.9 dB/mm baseline vs. 195.5 or 187.5 dB/mm, 3 h or 7 h EDTA, respectively N = 6) (Appendix A). These collective results were consistent with the primary echogenic surface at the CBI originating from the native or EDTA-displaced mineral front.

### 3.4. Qualitative Image Analysis

Transverse and segmented en face images of both the articular cartilage surface and CBI were created using custom-written MATLAB algorithms (Figure 5). Although the impact of brief EDTA decalcification was not clearly visible from the distance between the two horizontal lines in the transverse views corresponding to the articular cartilage and the recessed CBI/mineral front (Figure 5B1,C1), there was a clear increase in the distance between these two peak signals in A-mode line graphs (Figure 5F). This increased distance was significant and consistent across all briefly decalcified explants (Figure 4A,B). A non-echogenic “gap” in backscatter was seen in many explants in the cartilage deep zone immediately above the CBI (Figure 5, Appendix A).

Segmented en face images of the articular cartilage surface and CBI before and after EDTA decalcification allowed us to visualize the contribution of mineral deposits to the CBI backscatter. These images revealed that osteochondral explants have an irregular subchondral stiffness pattern that shifts after the calcified cartilage layer is demineralized (Figure 5A3–C3). This observation was also consistent for all explants analyzed (Appendix A). The depth-wise displacement of the mineral front was difficult to observe, however, when relying solely on the segmented CBI enface images that spanned from 0.2 mm above to 0.2 mm below the maximal CBI signal.

To visualize the spatial distribution of backscatter signal within the explant and how it changes following brief or total EDTA decalcification, 3D images were rendered using the Maximum Intensity Projection technique. By comparing the 3D images at baseline and post-EDTA treatment, the contribution of mineral density to backscatter was clearly visible by comparing both brief and long decalcified explants to baseline 3D images (Figure 6). The data showed a punctate rather than a smooth backscatter, which was unexpected given the relatively uniform histological mineral front and relatively flat tidemark line in histology sections (Figure 5D,E). Below the flat tidemark line, however, the mineralized tissue thickness undulated (Figure 5D,E, Appendix A). The most pronounced changes were observed in the totally decalcified explants, where a dramatic loss of backscatter occurred at the CBI (Figure 6 and Figure 7). Fully demineralized subchondral trabecular bone produced echogenic signals that penetrated up to 2 mm below the tidemark (Figure 7).

## 4. Discussion

### 4.1. Quantitative Ultrasound of CBI Remodeling

This study contributes a new and precise non-destructive quantitative imaging method that detects minor shifts in CBI mineralization in osteochondral explants. Brief EDTA treatment was used to mimic CBI remodeling and offers a unique novelty in our approach compared to previous studies. Collective data from this study suggest that the CBI backscatter peak originates from the tidemark mineral front in native tissue. The relatively smooth tidemark line did not generate an even backscatter intensity at the CBI. The structural basis behind the heterogeneous CBI backscatter strength remains unclear; one explanation is that highly echogenic patches arise from areas with a thicker contiguous mineralized tissue that shift upon brief surface-directed EDTA demineralization. This notion is supported by the loss of the CBI backscatter pattern after complete decalcification of the CBI (Figure 7). It is important to note that the explants used in this study were never exposed to chemical fixation, suggesting that live explants could be analyzed using the parameters optimized here. Future studies will use these imaging methods to monitor dynamic structural changes in long-term bovine explant cultures [33] supplemented with factors that could induce biological remodeling of the tidemark. The ultimate clinical utility of this work will arise when live explant cultures and non-invasive HFUS monitoring are used to identify new drugs that limit tidemark advancement that are then translated to clinical trials.

Our method has several limitations. We assumed a constant speed of sound (1600 m/s, native speed of sound in cartilage) when calculating HFUS cartilage thickness. The slight 7 µm discrepancy between HFUS and histological non-calcified cartilage thickness can likely be attributed to the varying speed of sound within the cartilage, which is influenced by factors such as glycosaminoglycan concentration and collagen fiber orientation [15,19,34]. We took into consideration that EDTA infiltration into cartilage could alter the speed of sound through cartilage, and took care to rinse the specimens in PBS before re-scanning to minimize the potential effect of salt on HFUS cartilage thickness measures. Steppacher et al. previously determined the native speed of sound through 1.38 mm thick bovine and porcine cartilage as 1580 m/s with an adjusted speed of sound as 1696 m/s using a 20 MHz transducer [20]. They attributed the adjusted speed of sound to the use of a transmission gel between the transducer and cartilage surface. In our study, three days of incubation in PBS at room temperature led to a significant 0.1 mm increase in HFUS cartilage thickness (Figure 4B). It is therefore possible that the speed of sound estimates in the Steppacher study were partly biased by storage of their samples in PBS before micro-CT imaging for ground truth cartilage thickness estimates [20]. Likewise, live cartilage could potentially undergo slight swelling in long-term explant cultures, and this would need to be addressed in future studies using longitudinal HFUS scanning to measure CBI remodeling. We concede that the speed of sound through cartilage could vary from one joint to another and also between different ages and species. It is noteworthy that clinical ultrasound systems assume a constant speed of sound. In future live explant cultures, the use of a matched pair design to analyze changes in HFUS cartilage thickness over time could partly address the limitation of an unknown specific speed of sound in each explant, assuming that the speed of sound is relatively stable in a given explant where the collagen network remains intact. Although the tidemark line was not distinctly visible in our histology images, HFUS cartilage thickness measurements consistently increased following brief EDTA treatment, suggesting that HFUS can detect changes in the mineralization state of the calcified cartilage layer that are not easily discernible through traditional histological analysis. The fact that we obtained such a high correlation between HFUS cartilage thickness and histological cartilage thickness for explants collected from the middle and periphery of the shoulder joints suggests that the difference in thickness between explants had a negligible effect on the results. These findings validate the use of HFUS cartilage thickness as a reliable QUS parameter for detecting mineralization changes.

The CBI backscatter signal was strongest when the acoustic focal plane was trained at the mid-zone or CBI. For explants imaged at the CBI focal plane, adult bovine explants showed a progressive 53 or 96 µm change in HFUS cartilage thickness, after 3 h or 7 h of EDTA, respectively. It is important to recognize that decalcification volume may vary between joints, even within the same species, due to factors such as age, sex, skeletal maturity, and prior injuries, which can influence the baseline mineral content and thickness of the calcified cartilage layer [34,35,36].

A strong correlation (R^2^ = 0.97) was observed between HFUS and histologically measured cartilage thickness using the von Kossa stain for phosphate mineral in non-decalcified histology sections. However, the exact volume of decalcification was difficult to determine from histology alone due to the inability to visualize the tidemark in hematoxylin-stained cryosections and analysis of only one section from the middle of each explant. Previous studies, such as those by Myers et al. [36], obtained good correlation of OA cartilage thickness between ultrasonic and histological measurements (R^2^ = 0.76). Another study using micro-CT validated the accuracy and reproducibility of A-mode ultrasonography for measuring cartilage thickness, reporting an excellent correlation (R^2^ = 0.95) and a mean accuracy of 74 µm [20]. Several clinical ultrasound imaging studies used 5 MHz to 15 MHz transducers to measure cartilage thickness [21,37,38], but these frequencies, according to our study, have insufficient resolution to diagnose tidemark remodeling. In addition, our data suggest that tidemark remodeling detection requires specimens with flat surfaces nearly perfectly perpendicular to the scanning axis, which is difficult to obtain when scanning larger areas with curved articular surfaces. It was also essential to use fresh unfixed specimens to avoid unknown effects of formalin cross-linking and to faithfully control specimen orientation, which we achieved in this study with custom 3D-printed holders and an immobilized support in the scanning bath.

### 4.2. Qualitative Analysis and Visualization

Qualitative analysis of the reconstructed images supported the quantitative findings. A clear gap in backscatter in the calcified cartilage zone was observed in post-EDTA transverse images, a feature that was consistently reproducible across all explants. The gap could be potentially due to the ability of sound to travel more readily through collagen fibers oriented along the same axis inside the radial zone above the tidemark [39]. The 3D images provided a more comprehensive spatial examination of the backscatter profile, making it easier to visualize changes between the baseline and post-EDTA. This qualitative assessment adds another layer of evidence supporting the sensitivity of HFUS in detecting the mineral front as the principle echogenic surface at the CBI.

## 5. Conclusions

To summarize, scanning methods optimized in this study provide a new imaging tool to study early mineralization changes at the CBI. These imaging methods could be helpful in detecting factors that initiate CBI remodeling toward identifying targeted treatments that halt OA progression, ultimately improving patient outcomes in OA management.

## Figures and Tables

**Figure 1 biomimetics-10-00160-f001:**
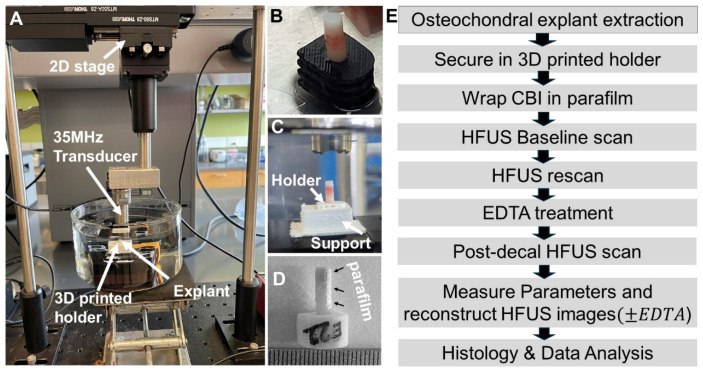
HFUS setup, sample holders, and final experimental workflow to measure incremental changes in cartilage thickness after EDTA-decalcification of bovine osteochondral explants. (**A**) HFUS setup showing the 2D motorized stage, 35 MHz transducer, 3D-printed support, and the osteochondral explant inserted in a 3D-printed holder submerged in a PBS-filled dish. (**B**) Custom U-shaped 3D-printed holder with explant inserted and (**C**) fit into the 3D-printed support. (**D**) The final protocol included parafilm wrapping of the explant CBI before the first baseline scan, followed by re-scan, and then EDTA decalcification and re-scan. (**E**) Flowchart detailing the final experimental workflow.

**Figure 2 biomimetics-10-00160-f002:**
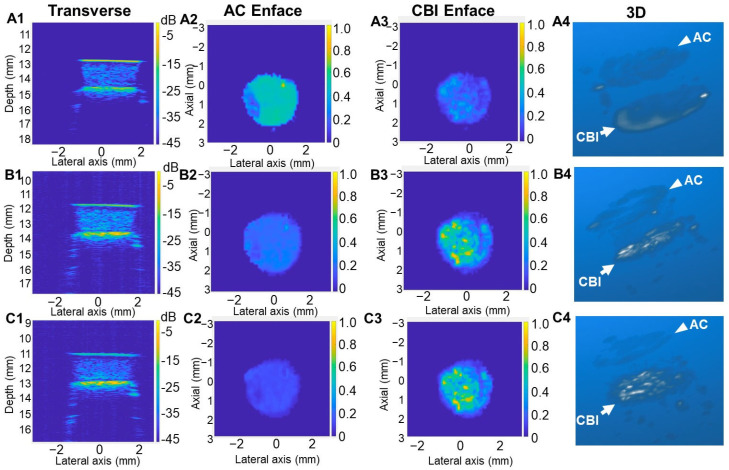
Effect of transducer focal distance on 35 MHz HFUS image quality. Images of a veal explant scanned with the transducer focal distance trained at the (**A1**–**A4**) articular cartilage surface, (**B1**–**B4**) mid-zone, or (**C1**–**C4**) CBI. Reconstructed images show (**A1**,**B1**,**C1**) transverse views, (**A2**,**B2**,**C2**) segmented en face articular cartilage surfaces (includes the first signal peak up to 0.2 mm below the first signal peak) or (**A3**,**B3**,**C3**) en face CBI (includes 0.2 mm above and 0.2 mm below second peak signal), and (**A4**,**B4**,**C4**) 3D reconstruction. Color scale (panels (**A2**–**C3**)): signal intensity (full-thickness signal intensity was normalized to a scale of 0 to 1 prior to segmentation). Abbreviations: AC: articular cartilage surface, CBI: cartilage bone interface, mm: millimeters, dB: decibels.

**Figure 3 biomimetics-10-00160-f003:**
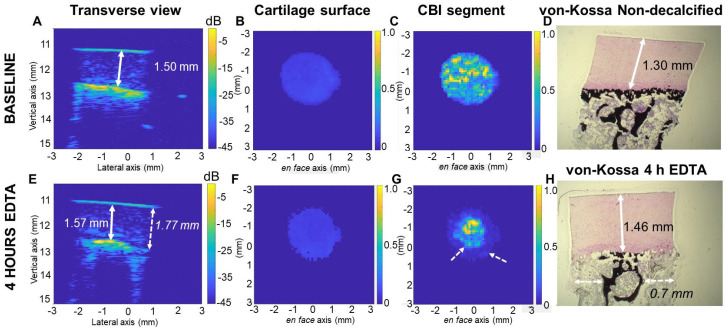
Brief EDTA treatment of osteochondral explants inserted in a 3D-printed holder with no other modifications resulted in top-down decalcification of the calcified cartilage layer but also outside-in decalcification that skewed the HFUS-cartilage thickness measures. (**A**–**C**) Transverse view of a baseline scan of a non-decalcified veal explant and (**D**) von Kossa (black mineral) and DMMB (pink stain) histology section of a different explant that was not treated with EDTA. (**E**–**H**) Transverse view of the explant shown in panel A after 7 h of EDTA decalcification and corresponding von Kossa histology of the same explant (**H**). HFUS cartilage thickness in a 1 × 1 mm ROI was ((**A**): 1.5 mm; (**E**): 1.77 mm) or ((**E**): 1.57 mm) in just the middle of the explant. (**B**,**F**) Segmented en face cartilage surface and (**C**,**G**) CBI. Dashed arrows (**E**,**G**,**H**) show outside-in decalcification. Color scale panels (**B**–**C**), (**F**–**G**): relative intensity. ((**D**,**H**) are shown at 2.5 magnification and are enlarged in Appendix A).

**Figure 4 biomimetics-10-00160-f004:**
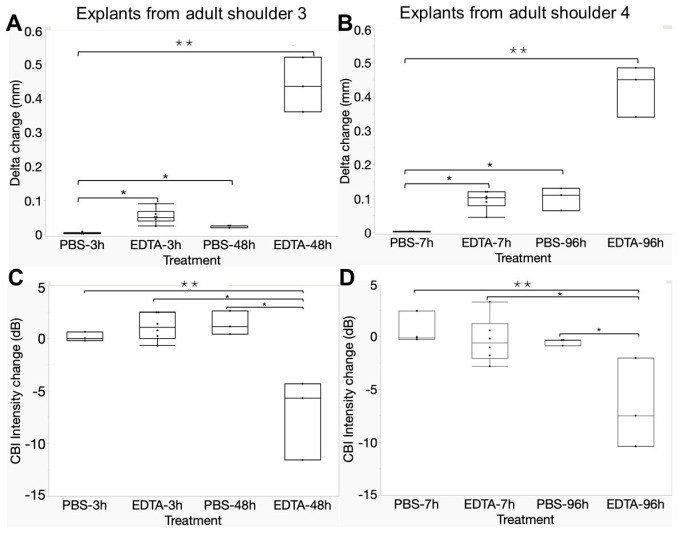
Incremental surface-directed EDTA decalcification led to an incremental increase in HFUS cartilage thickness without altering the CBI backscatter intensity. HFUS cartilage thickness was increased after brief (**A**): 3 h, (**B**): 7 h, and extensive (**A**): 48 h or (**B**): 96 h, surface-directed EDTA decalcification of adult bovine cartilage explants. Only long decalcification altered CBI backscatter intensity (**C**): 3 h, 48 h, or (**D**) 7 h, 96 h decalcification. Bovine shoulder 3 (**A**,**C**) or shoulder 4 (**B**,**D**). Values were calculated based on an average 674 ± 84 scan lines per explant. The graph shows median (horizontal line) and either interquartile range or min–max (box), min–max (whiskers), and individual data points (dots). *: *p* < 0.05; ** *p* < 0.01.

**Figure 5 biomimetics-10-00160-f005:**
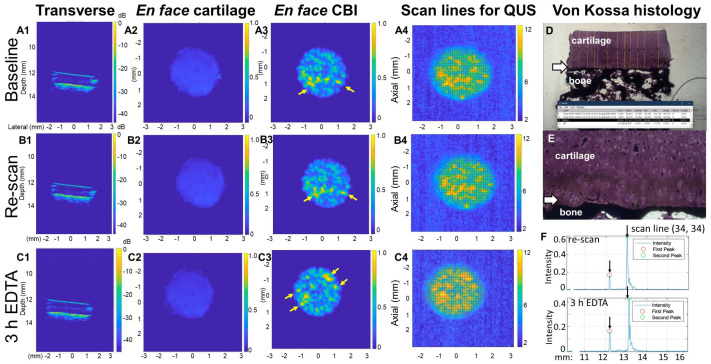
The inhomogeneous CBI backscatter pattern was reproducible in two baseline scans and shifted after brief surface-directed demineralization of the calcified cartilage layer. Panels show reconstructed HFUS images of an example explant at (**A**) baseline, (**B**) re-scan, and (**C**) post-3h EDTA. Panels (**D**,**E**) show post-3h EDTA von Kossa mineral stained histology where the white arrow points to the demineralized region, and (**F**) example raw scan line data (position 34, 34) before and after 3 h EDTA of the full-depth scan. Images show (**A1**–**C1**) transverse view, or en face segmented (**A2**–**C2**) articular cartilage surface, and (**A3**–**C3**) en face CBI. (**A4**–**C4**) En face image of the entire explant where dots indicate those scan lines used to generate QUS measures. Symbols: (**A3**–**C3**) yellow arrows—regions of strong CBI backscatter, (**D**,**E**) white arrowhead—region of demineralized calcified cartilage, horizontal yellow lines—line measures of non-calcified cartilage thickness, (**F**) vertical black line: position of the two backscatter peaks before EDTA treatment. Color scale (**A2**–**C2**,**A3**–**C3**): signal intensity, (**A4**–**C4**): signal intensity (volts). Full-thickness signal intensity (**A4**–**C4**) was normalized to a scale of 0 to 1 prior to segmentation into AC and CBI en face images (**A2**–**C3**).

**Figure 6 biomimetics-10-00160-f006:**
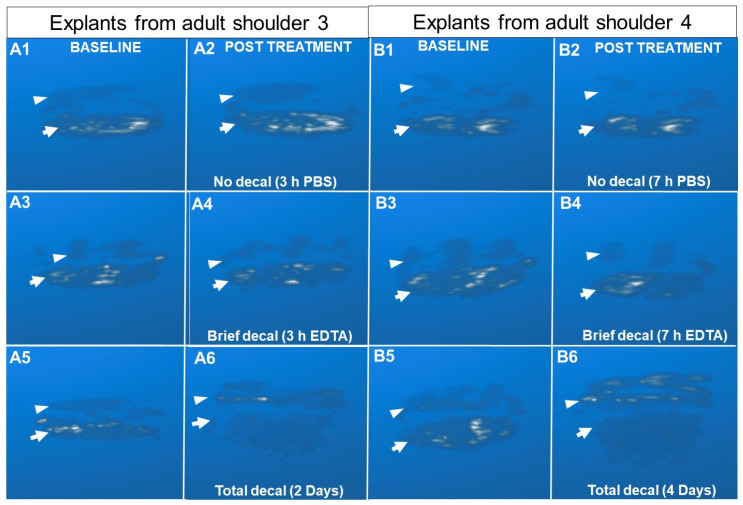
Reconstructed 3D images of explants before and after brief PBS (top row), brief EDTA (middle row), or total CBI decalcification (bottom row). (**A1**–**A6**) For adult shoulder 3, V4_E5 is in the top row, V4_E15 is in the middle row, and V4_E14 is in the bottom row. (**B1**–**B6**) For adult shoulder 4, AB4_E12 in the top row, AB4_E15 is in the middle row, and AB4_E5 is in the bottom row. Symbols: Arrowheads indicate the articular cartilage surface. Arrows indicate the CBI.

**Figure 7 biomimetics-10-00160-f007:**
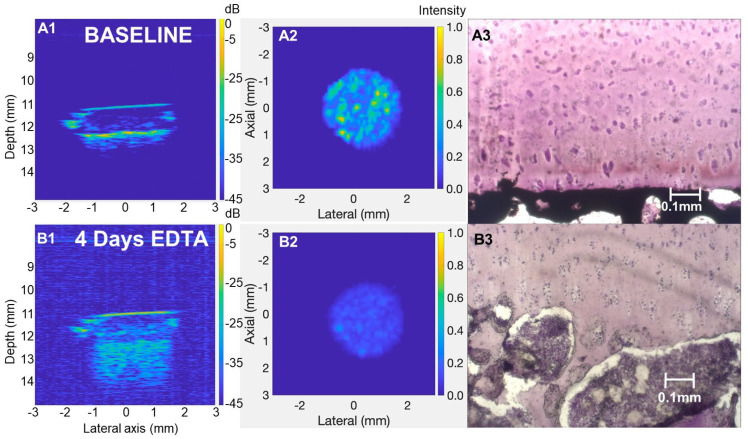
Full decalcification attenuated HFUS CBI backscatter. Panels (**A1**,**B1**) show transverse HFUS images of an adult bovine explant scanned at baseline and after 4 days of EDTA treatment, revealing changes in CBI backscatter intensity due to total mineral depletion. Panels (**A2**,**B2**) present corresponding segmented en face HFUS images of the CBI, comparing baseline and post-treatment backscatter profiles. Panel (**A3**) displays a 10× magnification histological section of a non-decalcified adult bovine explant, stained with von Kossa and hematoxylin, illustrating an intact CBI mineral front, while Panel (**B3**) shows a similar 10× magnification image of an explant after 4 days of EDTA treatment, demonstrating extensive mineral depletion extending into the subchondral bone.

**Table 1 biomimetics-10-00160-t001:** Study design.

Shoulder	Condition	# of HFUS Scans per Explant	*N*
Pilot study, 15 MHz transducer, effect of fixation, decalcification #			
Adult-1	Fixed, EDTA decalcified	5 (baseline, fixed, 1, 2, 4 days EDTA) 1 (±1 day EDTA)	6 2
Focal depth at the articular cartilage surface, mid-zone and CBI, 35 MHz transducer ##			
Veal	No treatment (scan in PBS)	6 (2× surface, 2× mid-way, 2× CBI)	4
Adult-2	No treatment (scan in PBS)	6 (2× surface, 2× mid-way, 2× CBI)	4
Demineralization study #1, 35 MHz transducer, CBI focal depth, ±parafilm wrap ##			
Veal	EDTA decalcification (1.5 h to 22 h)	2 (before and after EDTA or PBS)	5
Adult-2	EDTA decalcification time (3 h to 24 h)	2 (before and after EDTA or PBS)	5
Demineralization study #2, 35 MHz transducer, CBI focal depth, parafilm wrap ##			
Adult-3	EDTA brief decalcification (3 h)	3 (2 baseline, 1 post-EDTA)	6
Adult-3	Control: PBS incubation (3 h)	3 (2 baseline, 1 post-PBS)	3
Adult-3	Control: PBS incubation (48 h)	3 (2 baseline, 1 post-PBS)	3
Adult-3	Control: long decalcification (48 h)	3 (2 baseline, 1 post-EDTA)	3
Adult-4	EDTA brief decalcification (7 h)	3 (2 baseline, 1 post-EDTA)	6
Adult-4	Control: PBS incubation (7 h)	3 (2 baseline, 1 post-PBS)	3
Adult-4	Control: PBS incubation (96 h)	3 (2 baseline, 1 post-PBS)	3
Adult-4	Control: long decalcification (96 h)	3 (2 baseline, 1 post-EDTA)	3

# removed and re-inserted into 3D-printed holder between scans. ## kept in dedicated 3D-printed holder.

## Data Availability

Data and protocols will be made available by the corresponding author upon reasonable request.

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
