# Peer review of "Optimization of High-Frequency Ultrasound Imaging to Detect Incremental Changes in Mineral Content at the Cartilage–Bone Interface Ex Vivo"

_biomimetics, 2025, doi:10.3390/biomimetics10030160_

Round 1

Reviewer 1 Report

Comments and Suggestions for Authors

This manuscript describes the development of high-frequency ultrasound imaging to monitor changes in mineral content at the CBI. In fact, this is a very valuable contribution to osteoarthritis research, since there is an urgent need for non-destructive methods to study joint remodeling. This work contributes original insights to the literature by addressing certain gaps in the optimization of HFUS for cartilage imaging, and thus may bring a further step in the osteoarthritis diagnostic process. The comments below are listed for the authors' considerations.

1.     Introduction: This introduction clearly outlines the rationale, emphasizing the novelty of tidemark remodeling detection using HFUS. However, it lacks key references that discuss other noninvasive imaging techniques, including recent developments related to MRI.

2.     Methods: The methodology was sound and sufficiently detailed to allow reproduction. Technical rigor is demonstrated through the inclusion of custom MATLAB codes and 3D-printed holders. However:

• Ethical considerations over the sourcing of tissues are briefly discussed but can be expanded on.

           The justification for specific time points for EDTA treatment is unclear.

3.     Results:

The well-presented data, appropriate statistical analyses, and figures show the main findings. However:

           The figure legends lack sufficient detail for a standalone interpretation.

Discussion of the variability in cartilage thickness should be placed within a broader context of specimen preparation.

4.     Discussion: This discussion provides critical analyses of findings in the context of other studies. Conversely, claims that are not supported by strong evidence that HFUS can indeed emphasize "live" remodeling require further consolidation. A refinement could be in providing a more critical analysis of the limitations that emanate from this study, specifically the assumption of a fixed speed of sound.

5. Major Flaws

1. Clinical Relevance: The lack of in-depth discussion on the clinical implications of HFUS findings minimizes the impact of the manuscript.

2. Variability in Preparation: The variability in the thickness of the cartilage, and how that might affect the results, has not been well addressed.

6. Minor Flaws

1.         Typographic figure caption errors: spelling, irregular abbreviations, such as "dB" vs "decibels.”

2. Lack of clarity in methodological descriptions, such as incubation time for PBS.

Reviewer 2 Report

Comments and Suggestions for Authors

The innovation of this study is investigating high-frequency ultrasound imaging to detect the changes in mineral content at the cartilage-bone. After careful reading of this manuscript, I have the following comments:

(1) The study has innovation and is interesting to readers.

(2) The study is good designed and conducted.

(3) Data is well collected and presented.

(4) manuscript is good and logically written with good discussion.

(5) findings may have clinical applications.

Reviewer 3 Report

Comments and Suggestions for Authors

The topic of the paper is relevant. The results of experimental observations are described in this work accurately. I propose publishing the material with following remarks:

1.     In the abstract there are abbreviations without any explanations – HFUS, EDTA and PBS. And in 169 line - QUS. It is common to provide a transcript where the concept, method, or parameter is first introduced. Here the list is placed in the end of the paper. That is possible, but it is inconvenient. In 156 line – there is no explanation what is the TTL signal means.

2.     The 161 – 163 - "data acquisition had a sampling frequency of 500 MS/s within a 10-µs time window. The pulser-receiver settings were as follows: Energy: 4; Gain: 26 dB; Damping: 2; filter: 5-MHz high pass filter (HPF)." The sampling frequency could be in MHz. And there are no any units for energy and damping. May be to add or cut it. There is no information about angle aperture of the ultrasonic focused probe beam. And it is not clear why the HPF of 5-MHz was used in experiment with transducer of 35 MHz.

3.     In line 167, 173, 178 – “AC” – what does it mean. Articular cartilage surface peak? Perhaps the full name could be used first, and then the short one.

4.     Line 174 – “the slope reflecting an increase in backscatter intensity between 200 µm prior to the CBI peak and the CBI peak”. It is not clear.

5.     Line 180 – My suggestion is to explain of the Maximum Intensity Projection algorithm.

6.     Line 229-231 – “The three explants with flat surfaces had the same HFUS cartilage thickness at all 3 focal depths and a re-scan precision of 20 ± 21 µm. By comparison, 5 explants with “tilted” surfaces showed a drift in average HFUS cartilage thickness at the different focal depths and a 3-fold lower re-scan precision (69 ± 57 µm, Supp. Fig. S4).” I haven't understood what "a re-scan precision of 20 ± 21 μm" means. What is the reason for a 3-fold lower re-scan precision?

7.     Line 207 – All the chapter “3.1. Optimization of HFUS focal depth and sample preparation” is not so interesting. The result is basic. It was found that the best contrast is achieved with depth focus scanning. B/Z conception was published by Passmann, Ermert (1994) 150 MHz in vivo ultrasound of the skin: imaging techniques and signal processing procedures targeting homogeneous resolution. Proceedings of IEEE Ultrasonics Symposium ULTSYM-94. doi:10.1109/ultsym.1994.40190910.1109/ULTSYM.1994.401909.

In Line 415-419 authors indicate two papers [32], [33] where are the problem of focusing is discussed before. My suggestion is to cut the chapter and put the references and method description into chapter 2.4., not to the chapter with the results and discussion.

8.     Line 232-234 - “Based on these results, additional care was taken to extract explants with flat surfaces perpendicular to the scanning axis. Subsequent scans were carried out, focusing the transducer on the CBI.” It is also ordinary, routine task. We always take care to ensure that the US probe's axis is perpendicular to the sample surface during measurement.

9.     Line 237 -238 – “EDTA treatment of a veal explant (4 h) or adult explant (7 h) was sufficient to decalcify the calcified cartilage layer and demineralize ~100 – 150 µm of tissue according to histology measures.” I haven’t understanding about value 100-150 µm. In fig 3 delta equal 1.5 mm.

10.  Line 291 – “These findings suggested that the calcified cartilage layer lost its inherent acoustic impedance after being demineralized by brief EDTA”. Does it mean that sonic waves velocity has been changed? And how can this be taken into account when measuring the thickness of the cartilage layer?

The work is methodical. The paper hasn't no quantitative assessment of the degree of cartilage decalcification, depending on the EDTA exposure, which can probably be done based on the data obtained. In my opinion, the advantage of this work is that the experiment is described in detail with details. The novelty of this approach lies in the way in which model samples are prepared for remodeling calcifications. Publication could be published after revision.

Round 2

Reviewer 1 Report

Comments and Suggestions for Authors

No further comments to authors!

Reviewer 3 Report

Comments and Suggestions for Authors

Thank to author for detailed and interesting research.